# Counselling for Chronic Insomnia in Swiss Pharmacies: A Survey Study Based on Case Vignettes

**DOI:** 10.3390/pharmacy11030105

**Published:** 2023-06-16

**Authors:** Fanny Mulder, Dimitri Löwinger, Stephen P. Jenkinson, Estelle Kaiser, Tamara Scharf, Micheline Maire, Simone Duss, Claudio Bassetti, Raphaël Heinzer, Reto Auer, Carla Meyer-Massetti

**Affiliations:** 1Institute of Primary Health Care (BIHAM), University of Bern, 3012 Bern, Switzerland; fanny.mulder@students.unibe.ch (F.M.);; 2Graduate School of Health sciences GHS, University of Bern, 3012 Bern, Switzerland; 3Department of Internal Medicine, Cantonal Hospital of Zug, 6340 Baar, Switzerland; 4Interdisciplinary Sleep-Wake-Epilepsy-Center and Swiss Sleep House Bern, Inselspital—University Hospital of Bern, University of Bern, 3010 Bern, Switzerland; 5Service for Pneumology and Sleep Research Center (CIRS), Lausanne University Hospital (CHUV), 1011 Lausanne, Switzerland; 6Centre for Primary Care and Public Health (Unisanté), 1011 Lausanne, Switzerland; 7Clinical Pharmacology and Toxicology, Department of General Internal Medicine, Inselspital—University Hospital of Bern, 3010 Bern, Switzerland

**Keywords:** sleep, sleep disorder, insomnia, sleep hygiene, cognitive behavioural therapy, survey study

## Abstract

(1) Introduction: Chronic insomnia (CI) reduces quality of life and may trigger depression and cardiovascular diseases. The European Sleep Research Society recommends cognitive behavioural therapy (CBT-I) as the first-line treatment. Because a recent study in Switzerland demonstrated that this recommendation was inconsistently followed by primary care physicians, we hypothesised that pharmacists also deviate from these guidelines. The aim of this study is to describe current treatment practices for CI recommended by pharmacists in Switzerland, compare them to guidelines and examine their attitudes towards CBT-I. (2) Methods: A structured survey was sent to all the members of the Swiss Pharmacists Association, containing three clinical vignettes describing typical CI pharmacy clients. Treatments had to be prioritised. The prevalence of CI, and the pharmacists’ knowledge and interest in CBT-I were assessed. (3) Results: Of 1523 pharmacies, 123 pharmacists (8%) completed the survey. Despite large variations, valerian (96%), relaxation therapy (94%) and other phytotherapies (85%) were most recommended. Although most pharmacists did not know about CBT-I (72%) and only 10% had recommended it, most were very interested (64%) in education. Missing financial compensation hampers the recommendation of CBT-I. (4) Conclusions: Contrary to existing European guidelines, community pharmacists in Switzerland mostly recommended valerian, relaxation therapy and other phytotherapies for treating CI. This might be connected to the client’s expectation of pharmacy services, e.g., medication dispensing. While pharmacists recommend sleep hygiene regularly, most did not know of CBT-I as an overarching concept but were willing to learn. Future studies should test the effects of dedicated training about CI and changes in the financial compensation for counselling for CI in pharmacies.

## 1. Introduction

Chronic insomnia (CI) is a sleep disorder characterised by self-reported insufficient sleep quantity and quality. It is associated with symptoms such as difficulties falling asleep, maintaining sleep and waking early more than three times a week for more than three months [1].

In industrial countries, about a third of the population suffers from some symptoms of insomnia [2], and the prevalence of CI has been estimated to be about 10% [3]. In Switzerland, an interview study conducted in 2007 identified a prevalence of sleep problems that met the criteria for insomnia of 31.2% in the general population [4]. A more recent study analysed the data of 2432 patients cared for by 83 primary care physicians (PCPs) and reported similar findings, with 31% of patients reporting symptoms consistent with CI [5]. The prevalence of symptoms of insomnia reported by the clients of Switzerland’s pharmacies is even higher, at 40.9% among women and 38.7% among men [6].

CI is associated with psychiatric and medical conditions such as depression [7], cognitive impairment [8] and cardiovascular disease [9]. It can result in high direct and indirect costs (e.g., worse performance at work or an increased risk of accidents), and in 2015, these exceeded an estimated USD 100 billion in the USA alone [10]. Furthermore, insomnia may decrease quality of life, as was reported by 70% of the Swiss general population who reported having insomnia symptoms [4], contributing to insomnia’s importance as a public health issue.

In 2017, the European Sleep Research Society (ESRS) developed a set of guidelines for the treatment of CI, clearly stipulating that cognitive behavioural therapy for insomnia (CBT-I) was the first-line treatment [11]. CBT-I combines different elements, including relaxation methods, sleep hygiene, stimulus control, sleep restriction and other cognitive techniques of varying impact if applied as stand-alone options [12]. The use of sleep restriction alone has also proven efficient for improving insomnia severity [13,14]. The guidelines also state that medications such as Z-drugs or benzodiazepines should only be recommended as short-term treatments once CBT-I has been shown not to have worked. Other pharmacological (e.g., antihistamines, melatonin), phytopharmaceutical (e.g., valerian) and complementary treatments (e.g., homoeopathy) lack evidence of their efficacy, leading to a weak recommendation for their use [11].

A survey study of 3300 respondents in the USA revealed that a third of people reporting insomnia symptoms chose self-medication, with most opting for phytopharmaceuticals. That research found that 33% of its study population suffered from CI but that only 11% were consulting their physician about the symptoms [15].

This suggests that pharmacies could be an initial, low-threshold access point for people seeking help for their sleep problems. However, only limited evidence exists on the quality of pharmacists’ counselling for insomnia. One study from Jordan demonstrated, using a simulated patient, that pharmacists preferred conventional drugs and phytotherapy over CBT-I [16]. No data at all exists pertaining specifically to how pharmacists in Switzerland counsel their clients about CI. Information about pharmacists’ knowledge of CBT-I is also lacking. Indeed, a vignette-based survey study of PCPs in Switzerland demonstrated their limited knowledge about CBT-I as well as discrepancies between the treatments they initiated and current guidelines [17]. Consequently, our hypothesis derived from these findings was, that most pharmacists in Switzerland would advise their clients to use phytopharmaceuticals alone or in combination with a non-pharmacological treatment other than CBT-I.

We aimed to investigate the current clinical practice of Switzerland’s community pharmacists when counselling clients about CI, their estimation of CI’s prevalence among their clients, their knowledge of CI treatments and existing guidelines, as well as their willingness to learn more about them.

## 2. Materials and Methods

We conducted a structured online survey whose concept was based on a similar project aimed at PCPs and developed by Linder et al. [17]. The survey was adapted to community pharmacists and separated into four different parts, comprising items on demographic data about the pharmacists and their pharmacy, questions to be answered about three clinical vignettes, general questions about insomnia and questions about the pharmacist’s broad knowledge of CBT-I. The content was developed by an interprofessional team of pharmacists, PCPs, specialist physicians (neurologists, pneumologists), psychologists and clinical epidemiologists.

Clinical case vignettes have been shown to be a reliable, valid and inexpensive method of assessing clinical practice [18]. The first involved a 50-year-old female client with typical symptoms of insomnia but no medical diagnosis and wanting medication to treat her symptoms. The second involved a 45-year-old client, also with typical symptoms of insomnia, who had received a renewed prescription for three months of Zolpidem after a consultation with the PCP. The third vignette involved a 79-year-old client with the same story as the second client, the difference being that he was much older.

The original questionnaire in French was translated into German in collaboration with native German and French speakers. The complete questionnaire is available in English (validated using back-translation) in Appendix A.

The online questionnaire was managed using Findmind^®^, a Swiss online survey platform (www.findmind.ch, accessed on 13 May 2023, St. Gallen, Switzerland).

The survey was pilot tested with four community pharmacists. The preliminary version of the survey was distributed and each item answered by all pilot-testers. In addition, they were encouraged to give feedback on understandability, relevance and sequence of the questions. Feedback was reviewed among the research group and the survey adapted accordingly.

The survey was distributed electronically via an April 2021 newsletter sent to all the community pharmacy members of the Swiss Pharmacists Association, pharmaSuisse, at that time. Community pharmacies in Switzerland suffer increasingly from staff shortages. In order to increase geographically diverse participation and lower the burden of contributing to this extensive survey, we specifically requested one respondent per pharmacy. One month later, in May 2021, a reminder was sent out using the same means. Additionally, the investigators’ personal contacts at several cantonal pharmacists’ associations were encouraged to help increase response rates.

The invitation letter contained information about the study, the intent to publish anonymized data and an anonymised, non-personal link to the survey. Respondents could choose their preferred language (French or German) on the questionnaire’s webpage. We used Microsoft Excel^®^ (2016, Microsoft Corporation, Redmond, Washington DC, USA) for the descriptive analysis of the data. Continuous variables were reported as means and categorical variables as percentages. Incomplete surveys were excluded from the data analysis.

This study was carried out according to the principles of ethical research. According to the Human Research Coordination Office (KOFAM; www.kofam.ch, accessed on 13 May 2023), an ethics approval is unnecessary for irreversibly anonymized surveys. Participating pharmacists were informed in writing about the research project and the intended publication of anonymized data. By completing and returning the questionnaire, pharmacists voluntarily consented to their participation and the anonymous publication of the data collected and analysed within the scope of this study.

## 3. Results

Of the 1523 pharmacies contacted, 319 pharmacists participated in the survey, and 123 questionnaires were fully completed and could be included, making an overall response rate of 8% (*n* = 123/1523). Demographic data on the 123 participating pharmacists and their pharmacies are displayed in Table 1.

### 3.1. Knowledge about Insomnia

Pharmacists’ estimates of the prevalence of different types of insomnia varied widely. The majority (68%) estimated the prevalence of CI to be from 5–30% (37%, *n* = 45/120 estimated 5–15% and 31%, *n* = 37/120 estimated 15–30%). Details can be found in Figure 1.

### 3.2. Counselling and Dispensing Practices

When asked about the average time they needed to discuss sleep disorders (anamnesis and possible treatments) with their clients in the past, most pharmacists estimated from 5–15 min for sleep disorders in general (89.1%) and a maximum of 10 min for sleep hygiene (75.7%). Details are displayed in Table 2.

Box 1Case vignettes used to assess pharmacist counselling.Clinical case vignette 1 (CCV1) described a 50-year-old secretary who had developed sleep problems due to her situation at work. Falling asleep and waking during the night had been an issue for more than four months, affecting her performance. She asked for medication to help her sleep.Case vignette 2 (CCV2) described a 45-year-old cashier diagnosed with insomnia. He had taken Zolpidem for three months, returned to his PCP and received a repeat prescription, and had come to the pharmacy to have it filled.Case vignette 3 (CCV3) described similar circumstances as CCV2 in a much older patient (79 years) who had also taken Zolpidem for three months and had also come to the pharmacy with a repeat prescription which he/she wanted to have filled.

Figure 2 represents a comparison of the off-prescription pharmacological or homoeopathic treatments (including medications that the Swiss regulatory authorities allow pharmacists to prescribe autonomously, https://www.bag.admin.ch/bag/de/home/medizin-und-forschung/heilmittel/abgabe-von-arzneimitteln.html, accessed on 13 May 2023) that pharmacists in Switzerland would consider dispensing to clients in the different clinical vignettes.

Valerian was the most frequently chosen treatment for all three hypothetical clients (for a summary of the case vignettes, see Box 1). Almost all the pharmacists considered it for CCV1 (96%, *n* = 115/120), with 90% (*n* = 105/117) for CCV3 and 82% (*n* = 94/115) for CCV2. Other phytotherapies were the second most recommended treatment option for all three clients: CCV1 (85%, *n* = 97/114), CCV2 (81%, *n* = 91/112) and CCV3 (78%, *n* = 82/105). Regarding homoeopathic treatment, approximately half of the pharmacists considered it for all three hypothetical clients: CCV2 (54%, *n* = 57/106), CCV1 (50%, *n* = 55/110) and CCV3 (50%, *n* = 50/101). The choice of an antihistamine showed the most variation in the recommendations between the different clients, with more than half of the pharmacists considering it for CCV1 (53%, *n* = 61/115) and more than one in five for CCV2 (24%, *n* = 25/106) and CCV3 (21%, *n* = 22/103).

Figure 3 presents a breakdown of the medication that pharmacists would have suggested to a PCP in hypothetical interprofessional exchanges on the treatment of CI in each clinical case vignette.

The most commonly recommended medications were melatonin for CCV1 (70%, *n* = 77/110) and CCV3 (72%, *n* = 74/103) and an antidepressant for CCV2 (69%, *n* = 81/117), with, as a second choice, an antidepressant for CCV1 (58%, *n* = 65/113) and CCV3 (60%, *n* = 64/106) and melatonin for CCV2 (60%, *n* = 64/107). Z-drugs were the third most recommended choice for CCV1, suggested by 37% (*n* = 41/111) of pharmacists. However, for CCV2 with 28% (*n* = 30/106) and CCV3 with 36% (*n* = 38/106), the third most recommended choices were neuroleptics. Neuroleptics were the fourth choice for CCV1 (25%, *n* = 27/107), with Z-drugs being the fourth choice for CCV2 (17%, *n* = 18/105) and CCV3 (20%, *n* = 21/104). For all three hypothetical clients, the least considered choice was a benzodiazepine, with 17% (*n* = 19/109) for CCV1, 13% (*n* = 13/104) for CCV2 and 14% (*n* = 15/105) for CCV3.

A wide variety of non-pharmacological treatment options were recommended for the three case vignettes. Detailed results are available in Appendix A. However, when asked about elements of sleep hygiene that pharmacists would recommend to their clients in general, a certain trend was discernible: The most recommended sleep hygiene element, with 80% of the pharmacists recommending it, was not to use electronic devices before bedtime (82%, *n* = 97/119), followed by setting up a bedtime ritual (78%, *n* = 91/117). Additional details are displayed in Figure 4.

### 3.3. Knowledge about CBT-I

Three-quarters of our participating pharmacists had not heard of CBT-I (72%, *n* = 85/118) before answering the survey. Of the 29 pharmacists who had heard of it before completing the survey, more than a third had never recommended CBT-I to one of their clients (38%, *n* = 11/29). Thus, only 10% (11/118) sometimes recommended CBT-I. However, of the 29 respondents with knowledge about CBT-I, 25 (86%) did not know a psychologist or a CBT-I specialist to whom they could refer their clients. Twenty-seven of them (93%) did not know of a website or an application about CBT-I they could refer their clients to—only 2 (7%) did.

Participating pharmacists also had to indicate their interest in following up with a client doing an online CBT-I course over several weeks in order to help them complete the therapy. Overall, 76% (*n* = 89/118) of pharmacists were moderately or very interested in following up with their clients. When asked about their interest in attending courses addressing insomnia and CBT-I, 87% (*n* = 103/118) were moderately or very interested in the possibility.

Asked about prerequisites for the adequate counselling of insomnia patients in community pharmacies, 118 participants provided 215 answers: more than a third of these were requests for more knowledge or education on the topic (34%, *n* = 74/215), and another third suggested some form of financial compensation for the time invested in counselling clients in order to encourage a discussion about their sleep problems (33%, *n* = 71/215). A fifth of the pharmacists stated that better communication with PCPs was a prerequisite (19%, *n* = 41/215).

## 4. Discussion

To the best of our knowledge, this is the first study to assess sleep counselling practices in community pharmacies using clinical case vignettes and to inquire specifically about CBT-I.

Pharmacists participating in the survey were predominantly female.

### 4.1. Knowledge about Insomnia

Estimates of participating pharmacists about the numbers of their clients suffering from CI varied widely. However, the majority estimated the prevalence of CI to be between 5 and 30%. This is lower than the 31–36% of patients with insomnia symptoms identified in primary care practices [4,5]; however, it is reasonable to assume that the prevalence of properly diagnosed CI is lower than the prevalence of patients with insomnia symptoms. In addition, information on the current prevalence of insomnia in Switzerland is limited. In any case, itis important for healthcare providers to not only know about the prevalence and signs of CI but also about characteristics differentiating CI from other sleep disorders.

Current data on medications dispensed for insomnia in Switzerland are limited: a study published in 2021 extrapolated the one-year prevalence for the general Swiss population was 8.1% for a benzodiazepine prescription, 3.5% for a z-drug prescription and 10.5% for an either/or prescription, and continuously increased with age [19]. A report published in 2020 based on health insurance claims stated that 34% of home care clients used benzodiazepine at least once, while 24.3% used it for at least 3 months. Short-acting benzodiazepines were the least prevalent [20]. Ranked the 14th and 15th most commonly referred agents in nursing homes in 2016 were lorazepam and zolpidem, respectively; two agents used for sleep disorders that are also considered potentially inappropriate. One in ten nursing home residents had at least one use of zolpidem in that year, and twice as many had a use of lorazepam, a medium-acting benzodiazepine [21]. The collection of more comprehensive data, also pertaining to the use of OTC medications for insomnia treatment, might be beneficial to assess counselling practices more quantitatively in the future.

### 4.2. Counselling and Dispensing Practices

Valerian, a phytopharmaceutical, was the most recommended treatment among the non-prescription options available, closely followed by other phytotherapeutic medications. As stated in the ESRS guidelines developed in 2017, phytopharmaceutical medications such as valerian are not highly recommended [11]. Herbal supplements lack rigorous scientific data showing its benefit for treating insomnia symptoms [22].The level of evidence regarding different over-the-counter products for treating occasional sleep disturbances was also investigated by Culpepper et al. in a systematic review. They found a lack of clinical evidence about valerian’s efficacy and safety [23]. Nevertheless, phytotherapeutics might represent a logical choice in public pharmacies, potentially aligning well with customers’ expectations (dispensing drugs) and in light of limited pharmaceutical options with a decent risk–benefit ratio.

Pharmacological treatments requiring a prescription were generally less recommended than other treatment options. Among the prescription choices recommended to PCPs, melatonin was most common, followed by an antidepressant. Responses varied widely, not only between case vignettes but also within them.

Elements of sleep hygiene were recommended by 80% of the participating pharmacists, especially eliminating screen time before bedtime (82%) and/or setting up a bedtime ritual (78%). Although elements of sleep hygiene represent a part of CBT-I [12], CBT-I itself, as a comprehensive concept, was unknown to 72% of the pharmacists before they answered the survey.

Overall, these answers suggested that recognised guidelines for the treatment of CI were not being followed by pharmacists across Switzerland participating in this survey study. The guidelines unambiguously state that CBT-I should be the therapy of choice for CI. In a similar study of primary care practices in Switzerland, only 1% of patients received CBT-I [5]. Our results were also very similar to those from another study of PCPs in Switzerland conducted by Linder et al. [17]. PCPs also deviated from the guidelines by predominantly recommending elements of good sleep hygiene or prescribing phytopharmaceuticals and not CBT-I. As with our pharmacists, Switzerland’s PCPs were also unfamiliar with CBT-I. Thus, optimising the treatment of CI in Switzerland and avoiding potentially ineffective or harmful therapies will require a transformation of care.

Increasingly, community pharmacies have a separate counselling room at their disposal, favouring consulting services. However, the majority of pharmacists had never or rarely used a separate counselling room with their clients in the context of insomnia counselling. The limited duration of their discussions about sleep disorders, mostly 5–15 min, with up to 10 min devoted to sleep hygiene, could have contributed to the practice of not using a separate counselling space. Comparative data remain limited: one study performed in Jordan reported an average of 2 min and a maximum of 4 min devoted to sleep counselling for simulated patients [16]. Additional knowledge pertaining to CBT-I and sleep hygiene counselling might empower pharmacists to counsel more comprehensively.

### 4.3. Knowledge about CBT-I

Community pharmacies have the advantage of being very easily accessible to patients [24,25], and they could initiate triage or treatment depending on the symptoms and severity of the sleep disorders reported to them, including referring patients to PCPs or specialist physicians, including psychologists. However, even among those participants who had already heard of CBT-I, about 90% did not know a psychologist or CBT-I specialist to whom they could refer their clients, showing the need for awareness raising and information sharing. In addition, the known limited availability of psychiatrists and psychologists, not least because of the increased demand due to the Corona pandemic, might have influenced pharmacists’ treatment decisions

Participating pharmacists mentioned several barriers hampering better pharmacy integration: interprofessional collaboration should be optimised, knowledge updated and financial incentives made available. Similar results were reported from a recent study conducted in Australia: although pharmacists were interested in expanding their sleep health or insomnia practice, participants expressed the need for appropriate education or training, funding and collaborative treatment frameworks [26]. Non-existing reimbursement might contribute to the limitation of counselling services in general and also the perception of the availability of these services to the public, e.g., potential customers.

In Switzerland, improved financial compensation depends significantly on the authorities’ willingness to allocate new roles to pharmacists and on health insurance companies being prepared to pay for these new services, unless patients are willing to pay for some of them. Education, however, is very much in the hands of pharmacists themselves. Our study showed that 87% of participating pharmacists were moderately to extremely interested in following a course to improve their knowledge of insomnia and CBT-I. An important aid might be a CI decision-making tool for the use of Switzerland’s PCPs.

The present study had some limitations. While it covered most of Switzerland geographically, thanks to our survey’s availability in German and French, the participation rate was very low compared to another study in a similar setting [6], despite nationwide support from pharmaSuisse, which might limit its representability. Although it provided a comprehensive approach to reflect current practices, this four-part questionnaire was also time-consuming, which might have contributed to the limited response rate—much lower than the expected 30%. However, pharmacists in 23 of Switzerland’s 26 cantons participated in the survey, accounting for 88% of all cantons in total and their respective pharmacy practices. The distribution of the questionnaire during the Corona pandemic, with pharmacists heavily involved in testing and vaccination efforts, might also have contributed to the limited response rate. The low response rate prompted us to limit the interpretation of our results to a descriptive analysis and not perform subgroup analysis according to pharmacists’ education, experience and the location of their pharmacies.

In addition, because of the survey’s anonymity, we could not ensure that only one pharmacist per pharmacy answered our questions, potentially limiting the scope of our responses.

The survey introduction’s summary explaining CBT-I could have influenced some of the answers given and, therefore, introduced some bias.

The informational content of consulting practices is mostly limited to drug classes. Due to the extensive nature of the survey, we mostly did not inquire about specific active substances or recommended doses and regimens. This additional information would be very interesting to assess in a future survey.

For similar reasons and due to the questionnaire’s core topic of CBT-I, we explicitly did not ask about the reasons for current counselling practices. Therefore, the reasons for favourably recommending phytotherapeutics remains somewhat speculative. The structure of the answering options, separating potential answers into three groups (OTCs, prescription medications and non-pharmacological therapies), reflects the regulatory leeway pharmacists have in Switzerland. However, this limited the potential interlink between groups.

Future research should be undertaken to assess whether it is feasible for pharmacists to provide a counselling service for sleep disorders in general and CI specifically. Although the successful implementation of such a service in community pharmacies has been reported previously with brief behavioural interventions [27], a structured approach using a decision-making tool might facilitate pharmacist involvement and standardise treatment according to existing guidelines [28]. This might also facilitate a reimbursement fee for their time. One interesting option might be pharmacists facilitating the use of online tools [3,29,30,31], albeit their efficacy compared to face-to-face counselling needs more research.

## 5. Conclusions

The present study suggests that although many pharmacists counselled their clients about sleep disorders, their advice frequently deviated from that in existing guidelines as they favoured phytotherapeutic options over cognitive behavioural therapy for insomnia (CBT-I). The most proposed treatment was valerian, closely followed by other phytotherapies. Pharmacological treatments requiring a prescription were generally less recommended. There were large variations not only in the treatment types recommended by our participating pharmacists but also in their estimations of the prevalence of sleep disorders, acute insomnia and chronic insomnia (CI). While they were familiar with the different elements of good sleep hygiene, most pharmacists had not heard of CBT-I before completing the survey, and almost none of them had previously recommended it to a client. Nevertheless, most of them stated that they would be interested in taking a training course on the subject.

Our results suggest that changes in practice in Switzerland are necessary to make the treatment of CI compatible with current guidelines. Indeed, accessibility to care could be improved by more systematically involving pharmacists in screening and counselling for insomnia, as community pharmacies are easily accessible providers of primary care to the general population. This would require adequate interprofessional collaboration, together with training and financial incentives. There are many different levels of severity of insomnia and CI, requiring good communication between different types of healthcare professionals in order to provide individual patients with the best treatment possible. Including pharmacists in a step-by-step care approach could be an important contribution to making CBT-I more feasible and avoiding the use of potentially ineffective or harmful medication.

## Figures and Tables

**Figure 1 pharmacy-11-00105-f001:**
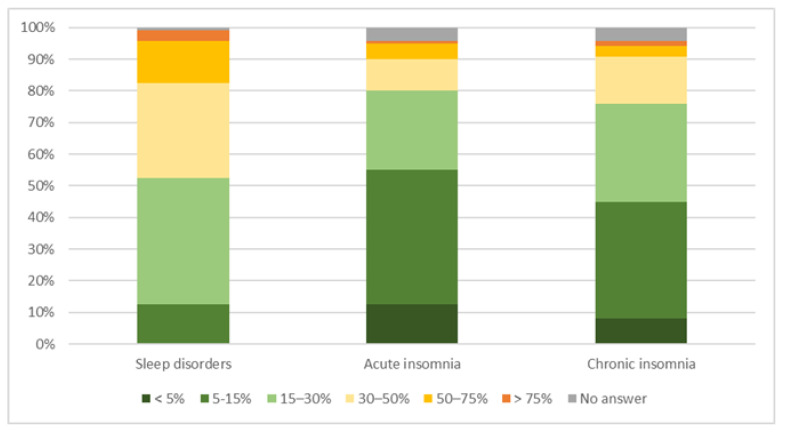
Estimated prevalence of community pharmacy clients suffering from sleep disorders, acute insomnia and CI.

**Figure 2 pharmacy-11-00105-f002:**
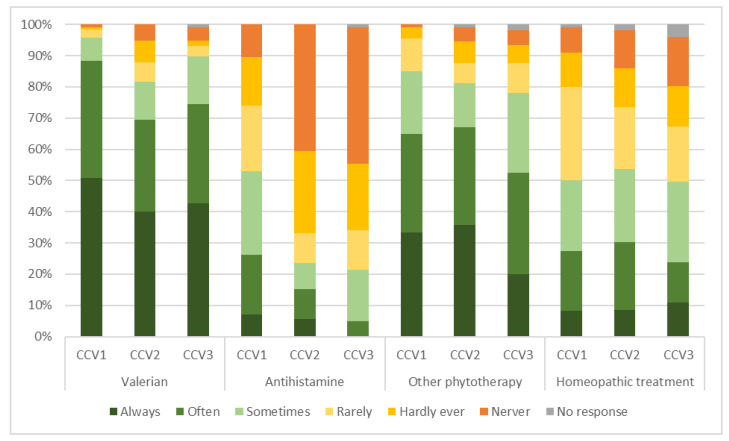
Overview of the non-prescription pharmacological treatment options recommended by community pharmacists in the scope of the three different clinical case vignettes. CCV = clinical case vignette.

**Figure 3 pharmacy-11-00105-f003:**
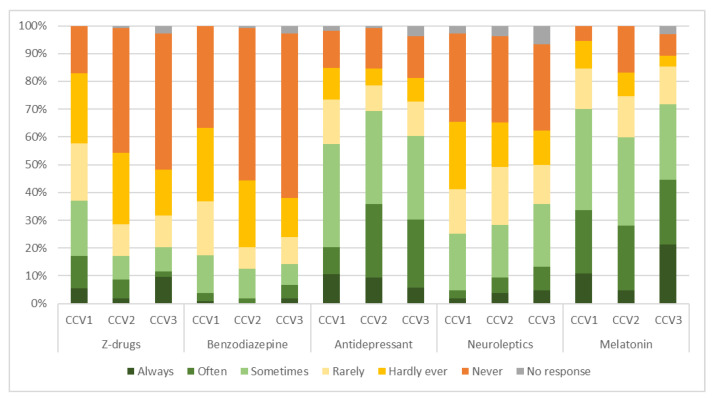
Overview of pharmacological prescription treatment options that community pharmacists would have suggested to a PCP for the three cases in the different clinical vignettes. CCV = clinical case vignette; PCP = primary care provider.

**Figure 4 pharmacy-11-00105-f004:**
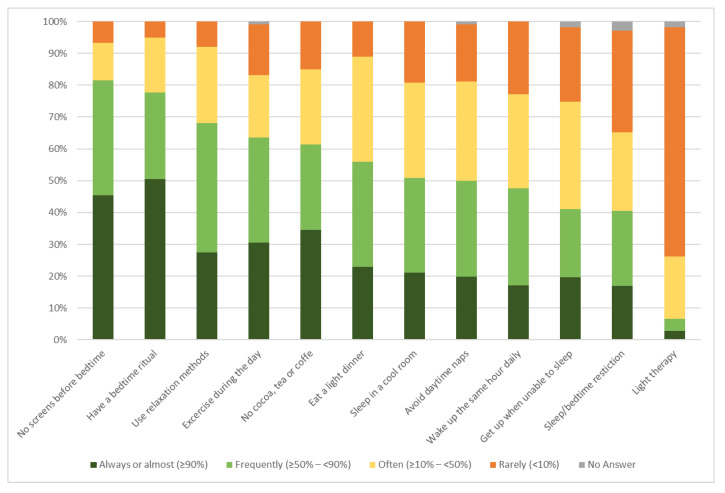
Overview of the sleep hygiene elements considered by community pharmacists for sleep disorders in general.

**Table 1 pharmacy-11-00105-t001:** Participants’ demographic data and community pharmacy data.

Characteristic (N = 123)	N (SD)	%
Sex		
Female	97	78.9%
Male	26	21.1%
Other	0	0.0%
Missing	0	0.0%
**Age group**		
<30 years old	12	9.8%
31–40 years old	20	16.3%
41–50 years old	26	21.1%
51–60 years old	40	32.5%
>60 years old	25	20.3%
Missing	0	0.0%
**Language**		
French	48	39.0%
German	75	61.0%
**Pharmacy experience in a community setting**		
<5 years	15	12.2%
6–15 years	27	22.0%
16–25 years	32	26.0%
26–35 years	28	22.8%
>35 years	21	17.1%
Missing	0	0.0%
**Function in the community pharmacy**		
Pharmacist and owner	19	15.5%
Managing pharmacist	28	22.8%
Deputy pharmacist	61	49.6%
Surrogate	10	8.1%
Other	5	4.1%
Missing	0	0.0%
**Initial university degree from**		
Basel (Switzerland)	36	29.3%
Bern (Switzerland)	10	8.1%
ETH Zurich (Switzerland)	27	22.0%
Geneva and/or Lausanne (Switzerland)	29	23.6%
Other	16	13.0%
Missing	5	4.0%
**Post-graduate pharmacy qualification**		
FPH in Community Pharmacy	47	27.0%
FPH in Hospital Pharmacy	0	0.0%
FPH in Clinical Pharmacy	0	0.0%
FPH in Anamnesis in primary care	36	20.7%
Course for leading quality circles	30	17.2%
CAS in Clinical Pharmacy, Geneva	1	0.6%
CAS in Clinical Pharmacy, Zurich/Basel	4	2.3%
Doctorate	9	5.2%
None	25	14.4%
Other	22	12.6%
**Location of the pharmacy**		
City centre	38	30.9%
Urban	47	38.2%
Rural	38	30.9%
Missing	0	0.0%
**Type of drug dispensing ***		
Dispensed by pharmacies only	52	42.2%
Dispensed by physicians only	42	34.1%
Mixed dispensing	28	22.8%
Missing	1	0.9%
**Community pharmacy structure**		
Pharmacy chain	29	23.6%
Pharmaceutical group	36	29.3%
Group-purchasing pharmacy	10	8.1%
Independent pharmacy	46	37.4%
Other	1	0.8%
Missing	1	0.8%

CAS = certificate of advanced studies (university-based); FPH = specialisation title for a national continuing education qualification. * In Switzerland, several cantons allow medication dispensing by practicing physicians directly from their practice stock to their patients. In this case, there is no pharmacy involvement.

**Table 2 pharmacy-11-00105-t002:** Average time spent discussing sleep disorders and sleep hygiene in community pharmacies.

	Discussing Sleep Disorders	Discussing Sleep Hygiene
<5 min	3.4%	32.8%
5–10 min	52.1%	42.9%
>10 to 15 min	37.0%	19.3%
>15 min	6.7%	5.0%
No answer	0.8%	0.0%

## Data Availability

All data supporting the findings of this study are available within the paper and its Appendix A.

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
