# Peer review of "Counselling for Chronic Insomnia in Swiss Pharmacies: A Survey Study Based on Case Vignettes"

_pharmacy, 2023, doi:10.3390/pharmacy11030105_

Round 1
Reviewer 1 Report
Review comments for:
“Counselling for chronic insomnia in Swiss pharmacies: a survey study based on case vignettes”
Journal Pharmacy (ISSN 2226-4787)
Manuscript ID pharmacy-2423985
Some general points to be addressed:
1- The Swiss Pharmacists Association has pharmacists that are mostly community-based or is it also health-system, including hospital, etc. (as the focus was data from community pharmacists)
2- Were specific agents listed or just categories (melatonin is specifically listed as is valerian but then general categories are listed for: antihistamine, other phytotherapy, homeopathic treatment, Z-drugs, benzodiazepine, antidepressant, neuroleptic(s). Please explain why some had categories listed and others mentioned specific agents.
3- What about other agents? (eg. lemon balm (or melissa), passionflower, chamomile, lavender, skullcap, etc. (may want to discuss this too as some patients prefer herbals with CBT-I as opposed to prescription agents. (also dd pharmacists have a place to list other alternatives (ie. Were agents only listed or could they fill in their own recommendations)
4- Would like to see more comparison data from the PCP and pharmacist recommendations.
5- Why was only 1 pharmacist per community allowed to reply? Since a more thorough reply could be generated from another pharmacist
6- May provide more discussion as to the data collected on the responders (the age group 51 o 60 years old and the > 60 years old group had higher response rate (could this be due to their own experiences?)
7- Would like to have seen question on number of patients they see in practice with insomnia as well as the prescriptions they fill for insomnia.
8- Need more data on WHICH antihistamine, which other phytotherapy, which homeopathic treatment, which Z-drugs, which benzodiazepine, which antidepressant, which neuroleptic, etc. which melatonin (as some are OTC and prescription) was preferred (the more specific the better)…this is important as big difference for insomnia with triazolam vs. flurazepam, etc.
9- Data for the CBT-I could be mentioned first (on page 6 and not page 8
10- With regards to melatonin any specific formulations? Even Rx vs.OTC?
11- For what was light therapy recommended for? Advanced sleep phase disorder?
12- Did they inform patients that they can access CBT-I as apps and online?
13- Would like more info on products recommended (e.g doses, dosage forms, etc.)
14- Check reference #14 on line 269 (page 9) is it really #14 or #17? Please check all references for correctness.
15- Remove “poor” from line 294 (page 10)
16- Agree that some questions could have been eliminated while others could have been used to gather more info (e.g specific type of antidepressant, antihistamine, etc.)
Author Response
Dear Reviewer
Thank you very much for your time and expertise offered to improve our manuscript.
We have addressed all the issues raised carefully.
Adjustments based on the comments are implemented in track change mode directly in the revised manuscript.
We have addressed the general points in detail.
Thank you very much and kind regards. On behalf of all authors, Carla Meyer
- The Swiss Pharmacists Association has pharmacists that are mostly community-based or is it also health-system, including hospital, etc. (as the focus was data from community pharmacists)
Thank you very much for pointing this out. We have clarified this issue with specifying in the methods sections that we addressed community pharmacies specifically.
While pharmaSuisse’ members are indeed mostly community pharmacists, the association represents all pharmacists in Switzerland. - Were specific agents listed or just categories (melatonin is specifically listed as is valerian but then general categories are listed for: antihistamine, other phytotherapy, homeopathic treatment, Z-drugs, benzodiazepine, antidepressant, neuroleptic(s). Please explain why some had categories listed and others mentioned specific agents.
We are aware that the “flight height” of the active substances seems not uniform throughout the manuscript. However, due to the length of the survey, especially the scope of the case vignettes, we decided to a) focus of the survey on current counselling options in Switzerland (for melatonin, only one RX product is available, for the class of antihistamines, only two OTC options), b) inquire specifically about valerian as the phytotherapeutic option with the best evidence base, and c) limit inquiries about Rx because they are normally not dispensed without a prescription.
Because other readers might have similar concerns, we have added an explanation to the limitations section. - What about other agents? (eg. lemon balm (or melissa), passionflower, chamomile, lavender, skullcap, etc. (may want to discuss this too as some patients prefer herbals with CBT-I as opposed to prescription agents. (also dd pharmacists have a place to list other alternatives (ie. Were agents only listed or could they fill in their own recommendations)
For every question, the participants had the option to add notes. The questionnaires are made available as supplementary material to clarify this issue. - Would like to see more comparison data from the PCP and pharmacist recommendations.
Thank you for pointing this out. We consciously refrained from more comparisons because the right to dispense Rx vs. OTC drugs of the two professions is very different. In addition, the clients of a community pharmacy might have a different level of insomnia burden than patients in primary care practices. Therefore, we decided to solely focus on objectively comparable baselines: the compliance with guidelines and recommendations of CBT-I. - Why was only 1 pharmacist per community allowed to reply? Since a more thorough reply could be generated from another pharmacist
Community pharmacies in Switzerland acutely suffer from staff shortages. In order to increase diverse participation and lower the burden of contributing to this extensive survey, we specifically requested one respondent per pharmacy. This information has been added to the manuscript. - May provide more discussion as to the data collected on the responders (the age group 51 o 60 years old and the > 60 years old group had higher response rate (could this be due to their own experiences?)
Due to the limited response rate in combination with the request to limit responses to one pharmacist per pharmacy, our statistician as advised to refrain from interpreting population-based data and to just present it as demographic descriptive background information. - Would like to have seen question on number of patients they see in practice with insomnia as well as the prescriptions they fill for insomnia.
Thank you for your interest in additional information. Unfortunately, there is no common database in Switzerland reporting this data. However, we have added a section of health claims data about benzodiazepines and z-drugs and highlighted the need for better information on medication use to treat insomnia. - Need more data on WHICH antihistamine, which other phytotherapy, which homeopathic treatment, which Z-drugs, which benzodiazepine, which antidepressant, which neuroleptic, etc. which melatonin (as some are OTC and prescription) was preferred (the more specific the better)…this is important as big difference for insomnia with triazolam vs. flurazepam, etc.
Thank you for pointing out this important aspect. Pharmacist in Switzerland have no prescribing rights for Rx-drugs for treating insomnia. While they might suggest better therapeutic options to physicians in the scope of a prescription validation (for example due to the presence of a potentially inappropriate medication), they do not make the final therapeutic decision. Therefore, we limited the scopr of our already extensive study. - Data for the CBT-I could be mentioned first (on page 6 and not page 8
The sequence of the results presented represent the sequence of the questionnaire. We aimed for consistency and therefor ask to keep the sequence in order to have a logical flow throughout all sections of the manuscript. - With regards to melatonin any specific formulations? Even Rx vs.OTC?
Because only one melatonin (Rx) product is registered as a medication in Switzerland, not distinction was necessary in the questionnaire. - For what was light therapy recommended for? Advanced sleep phase disorder?
Light is sometimes used therapeutically when circadian disturbances underlie insomnia. We have not inquired about the evidence-based justification in the eyes of pharmacists, though. - Did they inform patients that they can access CBT-I as apps and online?
Thank you for this question. The study did not provide information about CBT-I to the participants before completion of the study. A follow-up project will address educational and resource needs of community pharmacists in Switzerland. - Would like more info on products recommended (e.g doses, dosage forms, etc.)
While we understand the interest in more detailed data, due to the extensive scope of the study, we did not inquire about details exceeding drug classes. However, we have added that it might be of interest to do a more in-depth study about counselling practices in the future. - Check reference #14 on line 269 (page 9) is it really #14 or #17? Please check all references for correctness.
Thank you for pointing this out – we have checked the references accordingly and corrected reference 14 on page 9. - Remove “poor” from line 294 (page 10)
“Poor” has been removed

Reviewer 2 Report
This manuscript aimed to investigate the present clinical community pharmacy practices for chronic insomnia among Switzerland's community, their prevalence of the condition among the patients/clients they provide services for, what this pharmacist community knows about CI treatments and guidelines, and how willing these pharmacists are to learn more about the treatments and guidelines. The work produced helps to address a gap in understanding the quality of pharmacists' counseling for insomnia, which is important given the pharmacist's low-threshold access point for a common condition. The authors achieve most of their aims, although their language in the discussion and conclusion could be tempered given limitations related to generalizability (8% response rate from a potentially biased sample). The overall paper is well organized and written, although could be improved for readability by adding headers related to each of the four aims in the Discussion section. particularly in the Results and Discussion section. The submission could also by describing any methodology that informed the pilot testing to adapt the survey. On the whole, I find this manuscript to be relevant and original. More content related to these issues is requested in the track change comments below. Please see the additional comments embedded in the manuscript.

Author Response
Dear Reviewer
Thank you very much for your time and expertise offered to improve our manuscript.
We have addressed all the issues raised carefully.
Adjustments based on the comments provided in the PDF version of the manuscript are implemented in track change mode directly in the revised manuscript.
Given the limited response rate, we have tampered the findings of the discussion and conclusions sections as recommended. We have expanded the explanations pertaining to the pilot testing of the questionnaire. While we kept the sequence of the results and discussion sections, we have added subtitles to improve the structure of the manuscript as requested. Additional references have also been provided where demanded.
Thank you very much for the opportunity to submit a thoroughly revised version of our work, on behalf of all authors, sincerely, Carla Meyer-Massetti
